# Effects of Hydroxypropyl and Lactate Esterified Glutinous Rice Starch on Wheat Starch Gel Construction

**DOI:** 10.3390/gels8110714

**Published:** 2022-11-04

**Authors:** Yongqiang Gong, Tingting Gu, Tiantian Zhang, Songnan Li, Zhenyu Yu, Mingming Zheng, Yaqing Xiao, Yibin Zhou

**Affiliations:** 1Anhui Engineering Laboratory for Agro-Products Processing, Food Processing Research Institute, School of Tea and Food Science & Technology, Anhui Agricultural University, Hefei 230036, China; 2Joint International Research Laboratory of Agriculture and Agri-Product Safety of the Ministry of Education of China, Institutes of Agricultural Science and Technology Development, Yangzhou University, Yangzhou 225009, China

**Keywords:** hydroxypropyl, lactate esterified, glutinous rice starch, wheat starch, dilute solution, gel properties

## Abstract

An investigation was conducted into the impacts of hydroxypropyl glutinous rice starch (HPGRS) and lactate-esterified glutinous rice starch (LAEGRS) on a dilute solution and gel properties of wheat starch (WS) at different proportions (0%, 1%, 3%, 5%, and 10%). The results of dilute solution viscosity showed that hydroxypropyl treatment of glutinous rice starch (GRS) could promote the extension of GRS chains, while lactate esterification led to the hydrophobic association of GRS chains, and the starch chains curled inward. Different HPGRS: WS and LAEGRS: WS ratios, *β* > 0 and ∆*b* > 0, showed HPGRS and LAEGRS produced attractive forces with WS and formed a uniform gel structure. Compared with WS gel, HPGRS, and LAEGRS could effectively delay the short-term aging of WS gels, and LAEGRS had a more significant effect. HPGRS increased the pasting viscosity, viscoelasticity, and springiness of WS gels, reduced the free water content, and established a tighter gel network structure, while LAEGRS had an opposite trend on WS. In conclusion, HPGRS was suitable for WS-based foods with stable gel network structure and high water retention requirements, and LAEGRS was suitable for WS-based foods with low viscosity and loose gel structure.

## 1. Introduction

Wheat is one of the most essential grains in the world. It is not only a staple food but also the main raw material for preparing bread, biscuits, cakes, and other foods [1]. Starch, the main ingredient in wheat flour, plays an influential role in the quality of wheat flour products. It is well known that different starch-based foods require different salient product features, and there are other unanticipated features that degrade the quality of the products [2]. In view of this phenomenon, it is feasible to add additives to starch-based foods to enhance certain characteristics or improve food quality. Recently, interest in finding improvements in the quality of wheat starch (WS) has increased due to increased consumer demand for wheat-related products. There has been considerable research into improving the properties of WS by adding additional substances. Simple physical mixing with other types of starch is a safe and straightforward way to improve the properties of WS. For example, purple potato starch can effectively enhance the gel network structure of WS and improve the ability to resist fission [3]. Blending polysaccharides with starch is also a safe and easy method, like Mesona chinensis polysaccharide, Laminaria japonica polysaccharides, tea polysaccharide, and carboxymethyl cellulose, etc. [4,5,6]. In addition, as an alternative type of natural starch, modified starch is frequently used in starch-based products to obtain desirable properties, as it may interact with natural starch and alter the gel properties [7].

Modified starch refers to starch treated by a certain method to improve the original characteristics of the starch or produce new characteristics to meet the needs of industrial production [8]. Modification methods include physical, chemical, and enzymatic methods. Natural starch molecules have a large number of available hydroxyl groups, which can react with some chemical groups under a series of conditions to achieve the purpose of modification [9]. Chemical modification consists primarily of etherification, esterification, cross-linking, oxidation, and acid hydrolysis. Hydroxypropyl starch is a kind of etherified starch widely used in food production. Hydroxypropyl starch is prepared by introducing a hydroxypropyl group into starch, which increases its solubility, shear resistance, paste clarity, freeze-thaw stability; hence, it is used as an adhesive and a thickener in food products [10]. Hydroxypropyl starch is widely used as a thickener and stabilizer for beverages, pastries, meat products, and other food products [11]. Esterified-modified starch, as another commonly used chemically modified starch, is obtained by replacing the alcohol hydroxyl group on the glucose unit structure of starch with an esterifying agent (organic acid or inorganic acid). Esterified starch produced by organic acid modification is commonly considered as resistant starch, like citrate starch ester, malate starch ester, acetate starch ester, etc. [12]. The hydroxyl group on the glucose unit of esterified starch is esterified, resulting in a great change in the properties of natural starch, which is widely used in food, pharmaceutical, textile, paper and other industries [13]. Lactic acid, as a food grade organic acid, is widely used in food preservation, seasoning, acid agent and health-care products [14]. The anhydride formed by the dehydration of lactic acid can become esterified with the hydroxyl group on starch under certain conditions and a few researchers have investigated the properties of starch after lactate esterification. In the previous study, modified corn starch with organic acid combined with humidity and heat, the content of resistant starch increased significantly, and the viscosity decreased significantly [15].

Dilute solution viscometry (DSV) provides an effective, fast, and inexpensive technique for studying polymer interactions, which is widely used to determine molecular weight and conformation of molecules. Irani et al. [16] investigated the intrinsic viscosity, molecular weight, shape factor, voluminosity, conformation, and coil overlap parameters of the canary seed starch and wheat starch by DSV. Heydari et al. [17] made a further exploration, and investigated the dilute solution properties of canary seed starch and wheat starch with different temperature, salt type and salt concentration. There is no study on the dilute solution viscosity properties of mixed starch.

Glutinous rice starch (GRS) limits application in widespread food industry; due to the low solubility, shear resistance and gel strength, it is of great significance to prepare modified starch from GRS [18]. In this study, hydroxypropyl glutinous rice starch (HPGRS) and lactate esterified glutinous rice starch (LAEGRS) were used to replace WS. Dilute solution viscosity properties, gelatinization properties, dynamic rheological properties, texture properties, water distribution and microstructure of HPGRS-WS and LAEGRS-WS gels with different amounts of modified starch were investigated. Mechanisms of HPGRS and LAEGRS affecting the formation of WS gels have also been investigated. This study serves as a reference for the application of HPGRS and LAEGRS to WS foods.

## 2. Results and Discussion

### 2.1. Analysis of Dilute Solution Viscosity Results

#### 2.1.1. Intrinsic Viscosity

As the concentration of mixed starch increased, the specific viscosity (*η_sp_*) increased and showed a linear relationship, thus making it feasible to determine the intrinsic viscosity ([*η*]) by extrapolating experimental data. The [*η*] is the *η_sp_* of polymer solution as its concentration approaches 0, which reflects the liquid flow of each molecule flow friction; The value of [*η*] does not change with the concentration and can supply a useful indicator of the hydrodynamic volume occupied by an individual macromolecule coil, which only depends on the dimensions of the polymer chains [19]. The [*η*] of WS, HPGRS-WS, and LAEGRS-WS are shown in Table 1. As the proportion of HPGRS and LAEGRS replacing WS increased, the [*η*] of HPGRS-WS and LAEGRS-WS both increased. The [*η*] of single HPGRS and LAEGRS were 0.579 and 0.526, respectively, much higher than WS (0.279). It was related to the fact that both HPGRS and LAEGRS were prepared from GRS, with only minor amounts of amylose, and amylopectin is more difficult to recrystallize than amylose during the retrogradation [20]. As a result, the two modified starches were more difficult to undergo conformational contraction than the WS. The [*η*] of the two modified starches prepared from the same raw material was also wholly different, [*η*]_HPGRS_ > [*η*]_GRS_ > [*η*]_LAEGRS_. Previous studies proved that hydroxypropyl weakened the internal structure of starch and hydroxypropyl groups was hydrophilic, which is conducive to starch chains stretched in water, increasing the volume of starch in an aqueous solution [10]. The esterification of starch can lead to the formation of hydrophobic association in the gelatinization process and the curling of molecular chains, which reduces the volume of starch in an aqueous solution [21].

#### 2.1.2. Huggins Constant

The Huggins constant (k) is an essential parameter showing the aggregation of macromolecules, reflecting the molecular structure of polymer coils and the degree of coil expansion [16,17]. The value of k for perfect solutions is in the range of 0.3 to 0.7, whereas for flexible macromolecules with expanding shapes in a suitable solvent, the magnitude of this parameter is in the range of 0.2 to 0.4. For dilute solution, when the k value is higher than 1, it would bolster the formation of aggregates. As shown in Table 1, in deionized water at 25 °C, the k value of both types of mixed starches was greater than 1, which was related to the recombination of starch chains to form insoluble aggregates [19]. Among them, the k value of single HPGRS and LAEGRS were 2.90 and 3.42, respectively, which were lower than that of single WS (6.77). Compared to the two modified starches, WS has the higher amylose content and easier to form an ordered aggregated structure. Thus, with the increase of the WS component replaced by HPGRS and LAEGRS, the k value of the two mixed starchy species shows a downward trend. The k value of the mixed starch with the same amount of two modified starches had little difference, while the k value of the single two modified starches had obvious difference. The k value of GRS after hydroxypropyl decreased from 3.11 to 2.90, and the k value of GRS after lactate esterification increased from 3.11 to 3.42, which was attributed to HPGRS having higher hydrophilicity and stronger flexibility in an aqueous solution, while the GRS after lactate esterification developed hydrophobic association and weakened mobility [19,22].

#### 2.1.3. Molecular Conformation

In the dilute solution, the molecular conformation of the polymer is determined by calculating the slope of the b parameter of the power-law model. A slope (the b parameter) greater than 1 indicates the random coil conformation, while a slope less than 1 is associated with the rod conformation [23]. The b value of WS, HPGRS-WS, and LAEGRS-WS samples were shown in Table 1. As the replacement of WS by HPGRS and LAEGRS increased from 1% to 10%, the b value of HPGRS-WS increased from 0.532 to 0.582 and that of LAEGRS-WS from 0.533 to 0.569. The b value of single WS, HPHRS, and LAEGRS was 0.527, 0.972, and 0.946, respectively, indicating the rod-like conformation. Mahdi Irani et al. [19] reported the b value of canary seed starch (CSS) and WS samples were calculated as 1.274–1.352 and 1.489 (DMSO was used as a solvent), respectively, CSS and WS can be well dissolved in a DMSO solvent, distributed in an irregular state. In this study, water was used as the solvent, starch was difficult to dissolve at 25 °C, the retrogradation of starch resulted in a rod-like conformation. The b value of 0.527, 0.972, and 0.946 for the pure WS, HPHRS, and LAEGRS, respectively, indicated a rod-like conformation. The two kinds of modified GRS chains were more disorderly due to the elevated amylopectin. Compared to pure GRS, the b value of HPGRS increased from 0.959 to 0.972 due to the elongation of the starch chains facilitated by the hydroxypropyl modification, while LAEGRS decreased from 0.959 to 0.946, due to the curl of some of the GRS chains which was caused by hydrophobic association [19].

#### 2.1.4. The Parameter ∆b and β

Based on the results above, ∆*b* developed by Krigbaum and Wall [24] and *β* proposed by Sun et al. [25] were calculated. Two parameters were used to determine whether the interaction force between different polymers is attractive or repulsive. Table 1 showed that at different HPGRS:WS and LAEGRS: WS ratios (HPGRS: WS = 1:99, 3:97, 5:95, 10:90; LAEGRS: WS = 1:99, 3:97, 5:95, 10:90), *β* > 0 and ∆*b* > 0, indicating that HPGRS and LAEGRS were attracted to WS, respectively, at the four ratios. It was thought that macromolecules tend to unfold to a degree that exposes more possible interaction sites. In this study, the [*η*] of HPGRS and LAEGRS is significantly higher than that of WS, which means that both HPGRS and LAEGRS had sufficient hydrogen bond binding sites to interact with WS during gelatinization, which was the major reason for the mutual attraction of HPGRS, LAEGRS with WS [17]. It was proved that HPGRS and LAEGRS could form homogeneous gels with WS through chains entanglement, but the specific differences in gel properties of the two mixed starches need to be further explored.

### 2.2. Pasting Properties

Table 2 showed pasting properties of HPGRS-WS and LAEGRS-WS at different additive amounts using an RVA. Peak viscosity represents the viscosity value when the starch granule expands to its maximum, which is mainly affected by particle swelling, amylose leaching, and the interaction between starches [26]. Final viscosity stands for the ability of the starch paste to form a viscous paste upon cooling. The peak viscosity and final viscosity of the HPGRS-WS increased with the HPGRS fraction, which is mainly caused by the interaction between the HPGRS and the WS, as well as by the elevated expansion force of HPGRS. The steep expansion force of the HPGRS originates from the strong binding of the hydroxypropyl group to water molecules. Moreover, the hydroxypropyl weakens the structure of GRS grains, makes water permeable to starch grains more easily, and promotes the swelling of starch in the process of gelation [11,27]. Peak viscosity and final viscosity of LAEGRS-WS decreased with the increase of the proportion of LAEGRS. This phenomenon was ascribed to the fact that the interaction between LAEGRS and WS was weaker than WS only, and the steric hindrance of the lactate group and hydrophobicity of the carbonyl group in the LAEGRS was not conducive to the formation of hydrogen bond between starch and water in the gelatinization process, it inhibited the extension of LAEGRS [28,29].

The breakdown is the difference between peak viscosity and final viscosity, which reflects the heat resistance and shear resistance of starch [27]. The breakdown of HPGRS-WS increased with the proportion of HPGRS, but not significantly, indicating a slight decrease in its thermal stability. The result was related to the swelling power of starch promoted by hydroxypropyl groups [30]. While LAEGRS had no significant effect on the breakdown of WS.

The setback is the difference between the final viscosity and trough viscosity, it is caused by the retrogradation of starch with the decrease of temperature during the determination of pasting characteristics, which can reflect the short-term aging characteristics of starch [1]. The retrogradation of the two different mixed systems decreased significantly with increasing HPGRS and LAEGRS content, possibly due to the use of the two modified stalks in the GRS as a feedstock preparation. Amylopectin is the main ingredient in GRS, as we all know amylopectin is more difficult to age than amylose, and the increase of amylopectin in the mixed system is the main reason for inhibiting retrogradation [31]. As can be seen from Table 2, the setback of HPGRS-WS decreased more significantly than that of LAEGRS-WS when the sum of the additive quantities of HPGRS and LAEGRS is 10 percent. HPGRS has a weaker grain structure, and the internal amylose is more likely to leach, resulting in a higher amylose content involved in the retrograde process in HPGRS-WS than in LAEGRS-WS.

### 2.3. Dynamic Rheological Properties

The dynamic rheological results of HPGRS-WS gels and LAEGRS-WS gels were shown in Figure 2. The G′ and G″ of HPGRS-WS gels and LAEGRS-WS gels all increased with increasing frequency, indicating a significant frequency dependence between HPGRS, LAEGRS, and WS. The addition of HPGRS and LAEGRS had a significant effect on the dynamic rheological properties of WS gel. In the two different systems, the G′ of all tested samples was greater than G″ over the entire frequency range studied, and no crossover point occurred, indicating that all starch gels showed solid characteristics and were typically weak gels [32].

HPGRS increased the G′ and G″ of WS gel and enhanced the viscoelasticity of WS; it is attributed to the introduction of hydrophilic groups to reduce the steric hindrance of HPGRS, which promotes the dispersion of HPGRS in an aqueous solution and the release of amylose, and HPGRS and WS produced more molecular linking regions and formed a stronger cross-linking network structure [11,26]. For LAEGRS-WS, the higher the proportion of LAEGRS, the smaller the G′ and G″ of LAEGRS-WS, indicating that LAEGRS weakened the gel network structure of the WS, It was attributed to the increased steric hindrance of LAEGRS due to hydrophobic association, which inhibited the dispersion of LAEGRS starch chains in aqueous solutions and the exposure of hydroxyl binding sites, and ultimately resulted in lower interaction strength between LAEGRS and WS than WS only [19]. The tanα of the two different systems with different content of modified starch was less than 1, which showed an elastic behaviour [22]. The tanα of WS gel decreased with the addition of HPGRS, and the tanα of WS gel increased with the addition of LAEGRS, which also proved that HPGRS enhanced the gel network structure of WS, while LAEGRS weakened the gel network structure of WS.

### 2.4. Texture Profile Analysis

The gel hardness and springiness of the HPGRS-WS and LAEGRS-WS gels were shown in Figure 3. Hardness is the gel strength of a starch gel under pressure, attributed to the reassociation of starch molecules. Hardness is an important index to evaluate the degree of starch retrogradation which is positively correlated with amylose contents according to Liu [33]. As we could see Figure 3a, the hardness of HPGRS-WS and LAEGRS-WS gels was significantly decreased compared to WS gel, indicating that HPGRS and LAEGRS could effectively delay the retrogradation of WS gel, which was associated with a reduction in the amylose content of the two mixed systems. In addition, the hardness of the LAEGRS-WS gel was significantly lower than that of the HPGRS-WS gel when the addition of the two modified starches was 5% and 10%, respectively. The anti-retrogradation effect of LAEGRS on WS gels was stronger than that of HPGRS, which is consistent with the trend of setbacks in the pasting properties.

The springiness of the test samples can be expressed by returning to their original height after the compression force is eliminated, which can characterize the strength of the starch gel network structure [34]. The springiness of HPGRS-WS increased with the addition of HPGRS content, which was contrary to the changing trend of hardness, indicating that HPGRS promoted the construction of WS gel, and the interaction between HPGRS and WS was stronger than that of WS only. In contrast, the springiness of WS was significantly reduced after the addition of LAEGRS, indicating that LAEGRS was not conducive to the construction of WS gel network. These results are consistent with the results of dynamic rheology properties.

### 2.5. LF-NMR

In this study, the spin-spin relaxation time T_2_ was used to represent the degree of freedom of water, the bonding state, and the degree of free movement, of which the shorter the T_2_ time, the better the combination of water with the matrix [35]. There are three kinds of water with different fluidity in the T_2_ of starch gel during the relaxation time of 1–10,000 ms. The water characterized by T_21_ has the weakest fluidity and is strongly bound water. The part characterized by T_22_ is divided into weakly bound water, and the part characterized by T_23_ has the strongest mobility and is free water. A_21_, A_22_, and A_23_ represent strongly bound water content, weakly bound water content, and free water content, respectively [7].

The water fluidity and distribution could be determined by LF-NMR to investigate the water migration patterns in WS gels with different HPGRS and LAEGRS additions. Table 3 showed the peak area of each water state in HPGRS-WS and LAEGRS-WS calculated by using CPMG sequences from LF-NMR. Theoretically, strongly bound water represents water in a crystalline structure [35]. It could be seen from Table 3 that the relaxation times T_21_, T_22_, and T_23_ all decreased significantly with the addition of HPGRS. The lower spin relaxation time represented the tight bonding of water molecules in the gel [7]. A_21_ and A_22_ of the HPGRS-WS gel rise, and A_23_ showed an upward trend with increasing HPGRS, indicating the addition of HPGRS can effectively increase the strong and weak binding waters of the mixed starch gel.

Based on the above results, HPGRS and WS interact via molecular entanglement to form a stronger gel network structure. Compared with single WS gel, HPGRS-WS gel increased the content of strongly bound water by interacting with more water molecules due to the action of hydroxypropyl groups, and HPGRS-WS mixed starch gel network captured more water and increased the content of weakly bound water, resulting in a decrease in the content of free water [30]. The fact that the A_21_ and T_21_ of LAEGRS-WS were of no significant difference with WS showed the unclear change of strongly bound water, which could be due to the counteracting effect between increased amylopectin content and the lactate group. With the addition of LAEGRS, T_22_ and T_23_ of LAEGRS-WS showed an upward trend, moreover, A_22_ decreased and A_23_ increased, proved that LAEGRS-WS gel had less weakly bound water and more free water. Although LAEGRS was prepared from GRS, the association of GRS occurred after lactate esterification and LAERS and WS fail to form a more stable gel network structure, which was not conducive to water retention in the gel [1].

### 2.6. SEM

The formation of the gel among the starches was mainly attributed to the cross-linking between the starch chains and the formation of molecular bond regions through hydrogen bonds [3]. The morphology of the mixed starch gels observed under SEM were shown in Figure 4. All freeze-dried gels exhibited a continuous network structure and formed a honeycomb structure with different pore characteristics. Compared with WS gel, the gel network structure of HPGRS-WS was more ordered and tighter, and the cavity size of the gel was much smaller, which was the most obvious when the addition amount was 10% (Figure 4B–E). The results indicated that HPGRS could form a more continuous and denser matrix with WS, helping to trap more water molecules, which is consistent with the results in the HPGRS-WS gel water distribution. While the WS gel mixed with LAEGRS showed a completely different trend from HPGRS-WS gel. In the dilute solution viscosity results, LAEGRS and WS attract each other, corresponding to the scanning electron microscopy results, where LAEGRS-WS remains a continuous network structure. With the partial replacement of WS by LAEGRS increased from 1% to 10%, it could be seen that the network structure of the LAEGRS-WS was more and more sparse, and the irregular pores were increased (Figure 4F–I). Molecular entanglements in mixed starch gels were reduced and gel network structures were weakened.

### 2.7. The Potential Formation Mechanism of HPGRS-WS Gel and LAEGRS-WS Gel

The formation of a starch gel network structure was the result of the action of amylose and amylopectin; the introduction of different groups on the starch chains may have different effects on the gel [7,36]. In this study, the gelation mechanisms of HPGRS and LAEGRS mixed with WS were summarized separately. Based on the above results, hydroxypropyl increased the hydrophilicity of GRS and weakened the internal structure of starch particles. As a result, the starch chains of HPGRS were able to fully expand during gelatinization, with an increased number of exposed chain-binding sites. Compared with GRS, the [*η*] and b value of HPGRS increased, and the K value decreased. As a result, HPGRS had more binding sites and more entanglement with WS occurred, leading to a stronger gel network structure. This manifests itself in increased pasting viscosity, gel viscoelasticity and springiness, a more compact microstructure, and the retention of more water molecules in the network structure of starch gel. As in LAEGRS, esterification leads to the hydrophobic association of GRS molecules. The LAEGRS chains were coiled inward, and the exposed sites of starch chains hydroxyl were considerably reduced in the gelatinization process, which resulted in the decrease of the [*η*] and b value and the increase of k value compared with GRS. Dilute solution viscosity results showed that LAEGRS and WS still exhibit mutual attraction, proving that the lactate esterification only caused parts of the GRS chains to curl inward and that the massive starch chains were still entangled with WS. LAEGRS was unable to construct a stronger gel structure with WS due to the reduced number of molecular chains binding sites. It appeared in the results that LAEGRS reduced the pasting viscosity, viscoelasticity, and springiness of LAEGRS-WS, the gel microstructure was looser, and the ability to trap water was reduced.

## 3. Conclusions

The addition of HPGRS and LAEGRS had a significant effect on the formation of the wheat starch gel. The results of dilute solution viscosity showed that hydroxypropyl treatment of glutinous rice starch could promote the extension of glutinous rice starch chain, while lactate esterification led to the hydrophobic association of glutinous rice starch chain, and the starch chain curled inward. However, HPGRS and LAEGRS both had a mutually attractive force with WS, forming a uniform gel structure. Both HPGRS and LAEGRS can effectively delay the short-term aging of WS gel, and LAEGRS has a more significant effect. HPGRS improved the peak viscosity, final viscosity, dynamic modulus, and gel elasticity of WS gels, reduced the free water content, and established a closer gel network structure. HPGRS had a significant effect of thickening and promoting the formation of WS gels, while LAEGRS showed an opposite trend of the formation of WS gels. LAEGRS is more suitable for foods with low viscosity and gel strength requirements.

## 4. Materials and Methods

### 4.1. Material

GRS was isolated by the alkali steeping method described by Wang et al. [37]. The primary components of GRS were moisture 9.56%, ash 0.30%, protein 0.61%, fiber 0.10%, fat 0.07%, and amylose 0.47%. Wheat flour was provided by Bengbu Brothers Grain and Oil Co., Ltd. (Bengbu, China). WS was prepared according to Ding et al. [3], and its primary components were moisture 8.76%, ash 0.48%, protein 0.31%, fiber 0.09%, fat 0.10%, and amylose 24.31%.

HPGRS was prepared according to the method described by Yang et al. [22]. The water bath time was 9 h, the temperature was 42 °C, and the concentration of propylene oxide was 10%. The starch was dried at 40 °C until it reached a constant weight and ground in a laboratory grinder, then passed through a 100-mesh sieve and stored in airtight containers.

LAEGRS was prepared according to Butt’s method [38] with some modification. Lactic acid (30 g) was dissolved in 100 mL of distilled water, and the pH of the solution was maintained to 3.5 using 10 M NaOH solution. 100 g of GRS with a lactic acid solution was mixed at 25 °C for 12 h, the mixed sample was dried at 60 °C until the moisture content was 5–10%. The sample was taken out and cooled to room temperature, then the sample was further dried at a set temperature (110 °C) for esterification for a certain time (120 min). When the reactive starch was cooled to room temperature, stirred with water several times, and pumped to remove unreacted lactic acid and polylactic acid self-polymers. The samples were dried at 40 °C until they reached a constant weight and ground in a laboratory grinder, then passed through a 100-mesh sieve and stored in airtight containers.

### 4.2. Sample Preparation

Samples were prepared by replacing WS with HPGRS and LAEGRS at 0%, 1%, 3%, 5%, and 10% levels, respectively. The samples were fully mixed in the stirrer, and the mixture was evenly mixed, transferred into an airtight plastic container, and stored with P_2_O_5_ (Aladdin Chemical Reagent Co., Ltd., Shanghai, China, 99.99%) until use.

### 4.3. Dilute Solution Viscometry (DSV)

DSV was performed using a Ubbelohde viscometer (Cannon Instrument, State College, PA, USA) according to Guo et al. [17] with some modifications. The prepared samples were gelatinized at 95 °C for 30 min under different concentration gradients (0.15, 0.2, 0.35, 0.5, 0.7 g/dL). The ternary system (polymer 1-solvent-polymer 2) was measured using a Ubbelohde viscometer. The viscometer was immersed in a constant-temperature tank and run at 25 °C ± 0.1 °C. The relative viscosity and specific viscosity were calculated using the following formulas:(1)$$\eta_{r}=\frac{t}{t_{0}}$$
(2)$$\eta_{sp}=\eta_{r}-1$$
where $\eta_{r}$ is the relative viscosity, *η_sp_* is the specific viscosity, *t* is the outflow time of the blend solution, and *t*_0_ is the outflow time of the pure solvent. The results of each sample in the experiment were the average value of three parallel measurements, and the error was ±0.1 s.

Krigbaum and Wall [24] proposed that there were other interactions between polymers in the polymer-polymer dilute solution system, such as van der Waals forces and hydrogen bonds except for hydromechanical interactions. When the polymer concentration is low, the classical Huggins equation for a single polymer solution can be obtained by ignoring the higher-order term of concentration:(3)$$\left(\frac{\eta_{sp}}{C}\right)=\left[\eta \right]+bC$$
where
(4)$$\left[\eta \right]=\underset{C\to 0}{\lim}\left(\frac{\eta_{sp}}{C}\right)$$
and
(5)$$b=k{\left[\eta \right]}^{2}$$

Specific viscosity *(η_sp_*) is the increased specific viscosity of polymer (*η_sp_* = *η_r_* − 1); [*η*] is the intrinsic viscosity of the solution; *C* is the mass fraction of polymer in solution; *b* is a constant related to Huggins coefficient *k*, reflecting the interaction between polymer and polymer in solution under a certain concentration condition. The *η_sp_* of blends can be expressed as the following equation (Subscripts 1, 2, and m correspond to polymer is 1, 2, and their mixture, respectively):(6)$$\eta_{sp,m}={\left[\eta \right]}_{1}C_{1}+{\left[\eta \right]}_{2}C_{2}+b_{11}C_{1}^{2}+2b_{12}C_{1}C_{2}+b_{22}C_{2}^{2}$$

Changing Equation (6) into Equation (3) form according to Equation (5):(7)$$\frac{\eta_{sp,m}}{C_{1}+C_{2}}={\left[\eta \right]}_{m}+b_{m}\left(C_{1}+C_{2}\right)$$

Here
(8)$$b_{m}=w_{1}^{2}b_{11}+2w_{1}w_{2}b_{12}+w_{2}^{2}b_{22}$$

Krigbaum and Wall [24] defined $b_{12}^{id}=\sqrt{b_{11}b_{22}}$ in the mixed solution under ideal conditions. Gravitational or repulsive forces between polymers can cause a positive or negative bias between $b_{12}$ and $b_{12}^{id}$, Pingping [39] proposed Δb to determine the attraction or repulsion between blends. The parameter Δb indicates the attractive (Δb ≥ 0) or repulsive (Δb < 0) interaction among polymer segments.
(9)$$\Delta b=b_{12}-b_{12}^{id}=b_{12}-\sqrt{b_{11}b_{22}}$$

The Huggins parameter $k_{m}$ will deviate from the ideal value when there is thermodynamic interaction between the polymer components. Define $k_{m}=k_{m}^{id}+\alpha$. Sun et al. [25] used parameter *α* to judge the compatibility between blends. The parameter *α* indicates the attractive (*α* ≥ 0) or repulsive (*α* < 0) interaction among polymer segments.
(10)$$\alpha =k_{m}-\frac{\left(k_{1}w_{1}{\left[\eta \right]}_{1}^{2}+2\sqrt{k_{1}k_{2}}w_{1}w_{2}{\left[\eta \right]}_{1}{\left[\eta \right]}_{2}+k_{2}w_{2}{\left[\eta \right]}_{2}^{2}\right)}{{\left(w_{1}{\left[\eta \right]}_{1}+w_{2}{\left[\eta \right]}_{2}\right)}^{2}}$$
where
(11)$$k_{m}=\frac{\left(k_{1}w_{1}{\left[\eta \right]}_{1}^{2}+2k_{12}w_{1}w_{2}{\left[\eta \right]}_{1}{\left[\eta \right]}_{2}+k_{2}w_{2}{\left[\eta \right]}_{2}^{2}\right)}{{\left(w_{1}{\left[\eta \right]}_{1}+w_{2}{\left[\eta \right]}_{2}\right)}^{2}}$$
(12)$$k_{1}=\frac{b_{1}}{{\left[\eta \right]}_{1}^{2}}$$
(13)$$k_{2}=\frac{b_{2}}{{\left[\eta \right]}_{2}^{2}}$$
(14)$$k_{12}=\frac{b_{12}}{{\left[\eta \right]}_{12}^{2}}$$

Jiang and Han [40] revised this criterion by defining another parameter *β*, as
(15)$$\beta =\frac{2\Delta kw_{1}w_{2}{\left[\eta \right]}_{1}{\left[\eta \right]}_{2}}{{\left(w_{1}{\left[\eta \right]}_{1}+w_{2}{\left[\eta \right]}_{2}\right)}^{2}}$$
(16)$$\Delta k=k_{12}-\sqrt{k_{1}k_{2}}$$

The parameter *β* is a function of w and Δ*k*; values of *β* ≥ 0 indicate attractive, whereas values of *β* < 0 indicate repulsive.

### 4.4. Pasting Properties

The pasting properties of mixed starch were determined using an RVA-TM Rapid Visco Analyzer (Perten instruments, Stockholm, Sweden) according to Yang et al. with some modifications [41]. The temperature changes during the test were as follows: the sample to be tested was kept at 50 °C for 1 min, heated from 50 °C to 95 °C within 4 min, kept at 95 °C for 2.5 min, then cooled to 50 °C within 4 min, and kept at 50 °C for 1.5 min. The peak viscosity, final viscosity, trough viscosity, setback, and breakdown were used to characterize the pasting properties. All measurements were made in triplicate.

### 4.5. Dynamic Rheological Properties

The DHR-1Rheometer (TA Instruments, New Castle, DE, USA) was used to investigate the viscoelastic properties of mixed starches as described by Yang et al. [22]. The linear viscoelastic range of samples was determined by strain scanning from 0.01% to 10% at 1 Hz. The frequency sweep was conducted from 0.1 to 10 Hz at 25 °C and 0.5% strain value with a parallel plate of 40 mm diameter and a 50-mm gap. The storage modulus (G′), loss modulus (G″), and loss tangent (tanα = G″/G′) are recorded as a function of frequency.

### 4.6. Gel Texture

The textural properties of starch gels were examined using a TA-XT plus texture analyzer (Stable Micro Systems, Surrey, UK). The starch paste formed by the RVA was transferred to an aluminum can (45 mm × 30 mm), quickly covered with an aluminum lid to avoid water loss, and refrigerated at 4 °C for 24 h. The experiment was carried out after the sample was raised to room temperature. Samples were compressed twice to 25% of the original sample height using P/0.5 R probe at the speed of 1.0 mm/s, to a distance of 10.0 mm, the trigger force was 5.0 g. Six repeated experiments were performed on the samples.

### 4.7. Low-Field Nuclear Magnetic Resonance (LF-NMR)

Water mobility of the mixed starch gels was determined on a MesoMR23-060H-I nuclear magnetic resonance (Shanghai Niumay Electronic Technology Co., Shanghai, China) according to the method of Ren et al. [42]. The samples (2.00 g) prepared by RVA were transferred to a 10 mm diameter NMR tube. The spin-spin relaxation time (T_2_) measurements were carried out by the Carr-Purcell-Meiboom-Gill (CPMG) sequence. The echo number of the CPMG sequence was set to 1024, the scan number was 8, and the time domain range of spin echo was set to 1000–2000 ms.

### 4.8. Scanning Electron Microscopy (SEM)

The samples prepared as described in Section 2.2 were made into a starch paste with a mass fraction of 10%. The gelatinization condition was 95 °C for 30 min. The starch paste was placed in a petri dish, cooled to room temperature, frozen at −18 °C for 24 h, and then taken out for freeze drying for 48 h. The cross-section of the central portion of the freeze-dried sample was taken. The sample was observed at 100× magnification under a Hitachi S-4800 scanning electron microscope (Hitachi, Tokyo, Japan). Samples were mounted and added to a double-sided sticky tape on an aluminum stub, coated with a thin film of platinum (10 nm), and then examined at an accelerating voltage of 3 kV.

### 4.9. Statistical Analysis

All numerical results represent the average of at least three independent replicates using SPSS 16.0 (SPSS Inc., Chicago, IL, USA). The analysis of variance using Tukey’s test at a significance level of *p* < 0.05. Figures were performed using Origin2019b (OriginLab Inc., Chicago, IL, USA).

## Figures and Tables

**Figure 1 gels-08-00714-f001:**
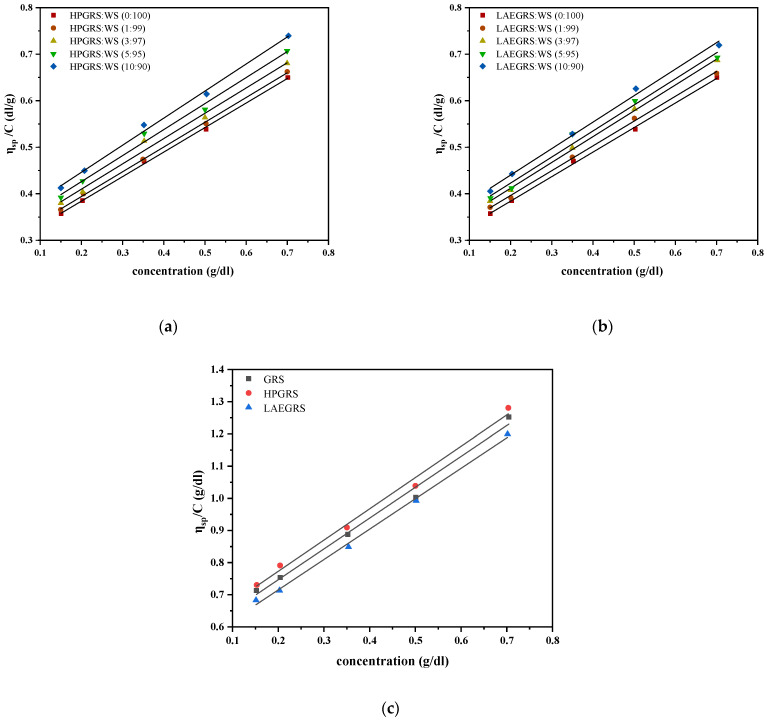
Reduced viscosity values of HPGRS-WS (**a**) at different ratios (HPGRS: WS = 1:99, 3:97, 5:95, 10:90); LAEGRS-WS (**b**) at different ratios (LAEGRS: WS = 1:99, 3:97, 5:95, 10:90); GRS, HPGRS, LAEGRS (**c**).

**Figure 2 gels-08-00714-f002:**
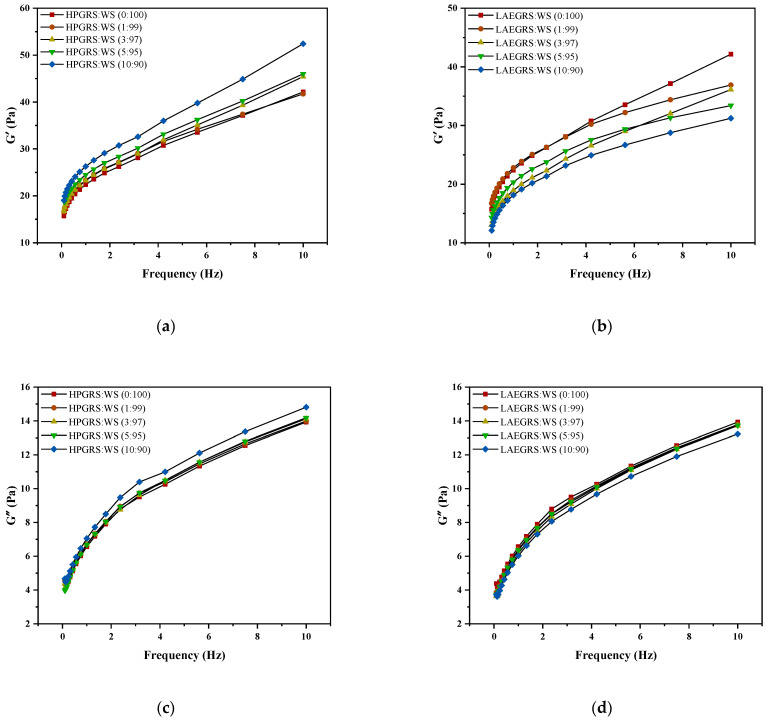
The revision is as follows: The G′ (**a**,**b**), G″ (**c**,**d**), and tanα (**e**,**f**) curves of different proportions of HPGRS and LAEGRS (0%, 1%, 3%, 5%, and 10%) blended with WS respectively.

**Figure 3 gels-08-00714-f003:**
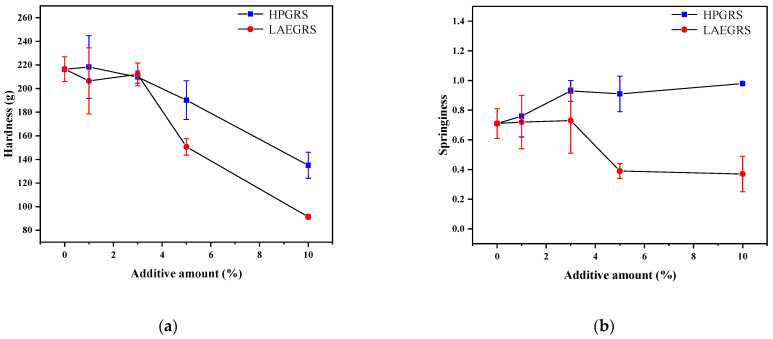
Hardness (**a**) and springiness (**b**) of different proportions of HPGRS and LAEGRS (0%, 1%, 3%, 5%, and 10%) blended with WS respectively.

**Figure 4 gels-08-00714-f004:**
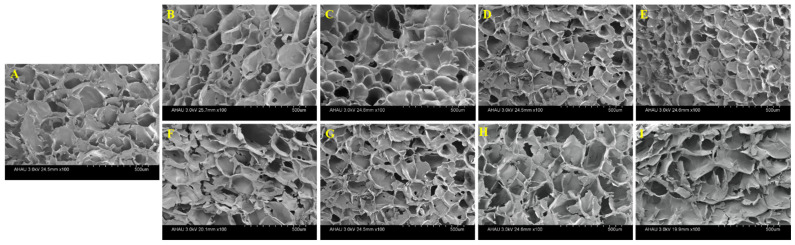
SEM images of different proportions of HPGRS and LAEGRS (0%, 1%, 3%, 5%, and 10%) blended with WS respectively.WS (**A**), HPGRS-WS (**B**–**E**), LAEGRS-WS (**F**–**I**).

**Table 1 gels-08-00714-t001:** The numerical values of polymer–polymer interaction coefficient and parameters for different proportions of HPGRS and LAEGRS (0%, 1%, 3%, 5%, and 10%) blended with WS respectively.

Samples	[*η*] (dL g^−1^)	*k*	*b* (dL g^−1^)^2^	Δ*b* (dL g^−1^)^2^	*β*
WS	0.279	6.77	0.527	-	-
HPGRS:WS (1:100)	0.288	6.41	0.532	0.06	0.08
HPGRS:WS (3:97)	0.301	6.00	0.544	0.10	0.18
HPGRS:WS (5:95)	0.314	5.67	0.559	0.14	0.23
HPGRS:WS (10:90)	0.330	5.34	0.582	0.09	0.30
LAEGRS:WS (1:100)	0.290	6.34	0.533	0.12	0.06
LAEGRS:WS (3:97)	0.301	6.13	0.555	0.30	0.14
LAEGRS:WS (5:95)	0.311	5.79	0.560	0.16	0.17
LAEGRS:WS (10:90)	0.326	5.35	0.569	0.03	0.17
GRS	0.555	3.11	0.959	-	-
HPGRS	0.579	2.90	0.972	-	-
LAEGRS	0.526	3.42	0.946	-	-

Note: [*η*] is the *η_sp_* of the polymer solution in Figure 1 when its concentration approaches 0; *b* is determined by calculating the slope of the power-law model in Figure 1; *k*, Δ*b*, *β* are calculated by the formula provided in Section 4.3, respectively.

**Table 2 gels-08-00714-t002:** Pasting properties of different proportions of HPGRS and LAEGRS (0%, 1%, 3%, 5%, and 10%) blended with WS respectively.

Samples	Peak Viscosity (cP)	Trough Viscosity (cP)	Final Viscosity (cP)	Breakdown (cP)	Setback (cP)
WS	3314.00 ± 141.27 ^bc^	2165.00 ± 41.58 ^a^	3260.67 ± 61.72 ^ab^	1149.00 ± 119.18 ^ab^	1095.67 ± 26.31 ^b^
HPGRS:WS (1:100)	3389.67 ± 50.52 ^bc^	2154.00 ± 3.21 ^a^	3279.00 ± 3.00 ^a^	1235.33 ± 51.21 ^ab^	1124.67 ± 2.08 ^a^
HPGRS:WS (3:100)	3459.00 ± 127.50 ^ab^	2205.00 ± 24.00 ^a^	3294.00 ± 27.00 ^a^	1254.50 ± 103.50 ^ab^	1089.00 ± 3.00 ^ab^
HPGRS:WS (5:100)	3460.00 ± 41.22 ^ab^	2217.67 ± 99.60 ^a^	3224.00 ± 46.87 ^ab^	1242.33 ± 58.45 ^ab^	1006.33 ± 68.07 ^abc^
HPGRS:WS (10:100)	3568.00 ± 10.54 ^a^	2245.00 ± 120.83 ^a^	3209.33 ± 30.11 ^ab^	1323.00 ± 114.24 ^a^	964.33 ± 124.03 ^bc^
LAEGRS:WS (1:100)	3317.60 ± 69.87 ^bc^	2143.67 ± 55.52 ^a^	3213.33 ± 77.53 ^ab^	1174.00 ± 15.72 ^ab^	1069.67 ± 22.28 ^ab^
LAEGRS:WS (3:100)	3333.50 ± 48.50 ^bc^	2247.50 ± 116.50 ^a^	3175.50 ± 17.50 ^bc^	1086.00 ± 68.00 ^b^	928.00 ± 99.00 ^cd^
LAEGRS:WS (5:100)	3349.00 ± 81.00 ^bc^	2238.50 ± 55.50 ^a^	3113.00 ± 54.00 ^c^	1110.50 ± 136.50 ^b^	874.50 ± 109.50 ^cd^
LAEGRS:WS (10:100)	3240.50 ± 45.50 ^c^	2095.00 ± 60.00 ^a^	2903.00 ± 19.00 ^d^	1145.50 ± 14.50 ^ab^	808.00 ± 41.00 ^d^

Note: Different letters in the same column indicate significant differences (*p* < 0.05), the same letters indicate no significant differences.

**Table 3 gels-08-00714-t003:** T_2_ relaxation distributions and factions of signal amplitude of protons of different proportions of HPGRS and LAEGRS (0%, 1%, 3%, 5%, and 10%) blended with WS respectively.

Samples	HPGRS-WS			LAEGRS-WS		
	T_21_ (ms)	T_22_ (ms)	T_23_ (ms)	A_21_ (%)	A_22_ (%)	A_23_ (%)
WS	1.73 ± 0.02 ^a^	24.13 ± 0.16 ^bc^	808.49 ± 7.26 ^b^	3.00 ± 0.19 ^b^	4.46 ± 0.06 ^b^	92.53 ± 0.13 ^a^
HPGRS:WS (1:100)	1.72 ± 0.01 ^ab^	24.19 ± 0.14 ^bc^	803.77 ± 6.36 ^b^	3.34 ± 0.36 ^ab^	4.18 ± 0.39 ^b^	92.48 ± 0.13 ^a^
HPGRS:WS (3:100)	1.70 ± 0.03 ^ab^	23.68 ± 0.24 ^cd^	748.58 ± 9.96 ^c^	3.31 ± 0.38 ^ab^	4.47 ± 0.37 ^b^	92.22 ± 0.21 ^a^
HPGRS:WS (5:100)	1.69 ± 0.01 ^ab^	23.31 ± 0.14 ^de^	735.03 ± 5.45 ^c^	3.43 ± 0.26 ^ab^	5.04 ± 0.04 ^a^	91.53 ± 0.22 ^b^
HPGRS:WS (10:100)	1.69 ± 0.01 ^b^	22.92 ± 0.20 ^e^	689.97 ± 1.45 ^d^	3.81 ± 0.08 ^a^	5.19 ± 0.08 ^a^	91.00 ± 0.09 ^c^
LAEGRS:WS (1:100)	1.73 ± 0.01 ^a^	24.09 ± 0.55 ^bc^	802.59 ± 6.92 ^b^	2.91 ± 0.21 ^a^	4.43 ± 0.06 ^ab^	92.66 ± 0.26 ^bc^
LAEGRS:WS (3:100)	1.72 ± 0.02 ^a^	24.62 ± 0.23 ^ab^	850.46 ± 24.02 ^a^	2.91 ± 0.39 ^a^	4.03 ± 0.28 ^abc^	93.05 ± 0.11 ^ab^
LAEGRS:WS (5:100)	1.72 ± 0.01 ^a^	24.91 ± 0.11 ^a^	851.40 ± 22.58 ^a^	2.85 ± 0.41 ^a^	3.76 ± 0.41 ^c^	93.39 ± 0.05 ^a^
LAEGRS:WS (10:100)	1.73 ± 0.01 ^a^	24.94 ± 0.16 ^a^	876.26 ± 28.97 ^a^	3.10 ± 0.23 ^a^	3.77 ± 0.17 ^c^	93.13 ± 0.27 ^ab^

Note: Different letters in the same column indicate significant differences (*p* < 0.05), the same letters indicate no significant differences.

## Data Availability

Data obtained as described.

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
