# Peer review of "Effects of Hydroxypropyl and Lactate Esterified Glutinous Rice Starch on Wheat Starch Gel Construction"

_gels, 2022, doi:10.3390/gels8110714_

Round 1

Reviewer 2 Report

The manuscript discussed the influence of blending wheat starch with hydroxypropyl and lactc ester of waxy rice starch. The introduction mostly leads well to the topic, however some information requires clarification. The aim of the work is clear. Materials and methods requires some extension (where there is citation of the method – basic principle should be provided). Discussion is mostly properly lead, nevertheless at several places it is speculative. There is no data included in the abstract.

 Detailed comments:

Line 49 not primarily, those 3 types are the only

Line 51 foreign (style)

Line 53-55 source should be provided, in Europe HP starch is not among the most commonly used preparations

Line 57 good additive? The used term is not clear

Line 63-67 esterified starch is generalization in this context, most of food grade modified starches are starch esters and each type of modification implies different changes to starch macromolecule. This was presented among others in https://www.mdpi.com/2227-9717/10/5/938/htm

Line 67-69 the reason that lactic acid or citric acid was not used extensively for starch modification is the fact that starch modification using those reagents is not considered / approved as food additives and cannot be used in production of food.

Line 69-71 in my opinion this information is not related and redundant

Line 72-74 the studies should be provided (cited)

Line 98 recombine?

Line 172 why interaction?

Line 189 please rephrase word power (as swelling power is tyle of starch pasete characterization)

Line 192-193 this depends on the employed program usually setback is correlated with decrease of starch paste temperature and thus we observe increase in viscosity. When the final temperature is low and determination time longer for example in Brandender apparatus, retrogradation may occure. Secondly thixotropic properties of the starch paste have also a minor impact on this value

Line 219 enhanced the viscoelasticity of WS – what is implied by that?

Fig 3 why only 2 TPA parameters were provided? How springiness was calculated? Usually is is a ratio and therefore its value is dimensionless.

Table 3 – since model starch system was analyzed (in fact two ingredients) why 3 proton fractions were identified? Moreover, how A values were calculated?

Line 363 – who short term aging was determined/defined?

Line 393- 429 very detailed description of rather simple experiment, please compact the information. It is also not clear for me why such simple apparatus (viscometer) was used when better alternative in form of rheometer was available?

Line 457 please provide the operation frequency of the equipment

 Author Response

Reviewer 3 Report

This study evaluates the effects of hydroxypropyl and lactate esterified glutinous rice starch on wheat starch gel construction. The article is interesting; however it needs some further clarifications.

In general authors should state the meaning of an abbreviation at its first use. In the introduction none of these are stated: WS, GRSs, HPGRS and LAEGRS. Also, in the intro authors must make it clear what is the importance of investigating the viscosity properties of dilute solutions.

I suggest author conduct a difference test in section 1 to determine how significant each substitution was.

In the discussion of frequency sweep I suggest authors include a hypothesis as to why G` increased with the increase in frequency. Also, it would be good to provide in the methodology what was the strain used for the frequency sweep as well as the amplitude results carried out to determine the linear viscoelastic region.

Why was a viscosimeter used instead of the rheometer? If the rheometer was used for both solution and gels maybe a correlation could be made.

Line 33: Such as? State some of the unanticipated features.

Line 34: remove “to correctly” as this is very subjective

Line 36: Define WS at first use

Lines 67-70: These sentences seem not to link with the content of the article. Why to emphasize lactate?

Line 93:Are shown

Line 237: Reological what?

Line 372: Why wasn’t fiber and fat (and consequently carbs) also determined?

Line 378: What was the basis for determining the temperature, time and concentration?

Line 379: By him or according to his method?

Line 381-383: Why is it written in imperative form?  Re-write

Line 391: P2O5? What is this?

Line 393: Provide further details of the equipment

Line 398: where

Line 466: refrigerated or frozen at -18 C?

Reviewer 4 Report

Thank you for inviting me to review this manuscript. My comments are as follows:

1. The sentence of the title “Effects of……: based on ……” looks strange.

2. More elaboration is needed to support the conclusion in line 72-73.

3. Physical quantity symbols are generally written in italics.

4. Shouldn't the pore structure characteristics of B and F in Fig. 4 be the same as those of A?

Round 2

Reviewer 1 Report

The revised article “Effects of hydroxypropyl and lactate esterified glutinous rice starch
on wheat starch gel construction: Based on dilute solution viscosity and gel
properties
” has been modified by reviewer’s comments and much improved. But, below information is still limited.

 -some instrument expressed as (company, city, state, country) and some expressed as (company, city, state, country) on Line 471, 472

Author Response

Our response: Thank you for the reviewers’ suggestion.

We changed all the instruments to the same format and expressed as (company, city, country) 

Reviewer 2 Report

The manuscript has been improved substantially. I have to remarks, regarding the changes in introduction.

Comment 7: the changes part is more detailed, but still concerns whole group of modification methods while they imply different effect

Comment 8 I think that removing this information was not the proper way, the information that was provided in the initial review should be included

Author Response

Comment 1:Comment 7: the changes part is more detailed, but still concerns whole group of modification methods while they imply different effect.

Thank you for the reviewers’ suggestion. We have corrected it. The revision is as follows: The hydroxyl group on the glucose unit of esterified starch is esterified, resulting in a great change in the properties of natural starch, which is widely used in food, pharmaceutical, textile, paper and other industries.

Comment 2:Comment 8 I think that removing this information was not the proper way, the information that was provided in the initial review should be included.

Thank you for the reviewers’ suggestion. We made some modifications, the revision is as follows: Lactic acid, as a kind of food grade organic acid, is widely used in food preservation, seasoning, acid agent and health care products. The anhydride formed by dehydration of lactic acid can esterified with the hydroxyl group on starch under certain conditions and a few researchers have investigated the properties of starch after lactate esterification.

Reviewer 3 Report

The authors have considerably improved the manuscript and addressed all the points that were raised. Just two other minor comments: Line 33: To add and Line 53: remove one point

Author Response

Thank you for the reviewers’ suggestion. The reviewer's suggestion has been accepted and we have corrected it.